# Does Educational Status Influence Parents’ Response to Bad News in the NICU?

**DOI:** 10.3390/children10111729

**Published:** 2023-10-25

**Authors:** Mirjam Wege, Pia von Blanckenburg, Rolf Felix Maier, Carola Seifart

**Affiliations:** 1Children’s Hospital, University Hospital, Philipps University of Marburg, 35033 Marburg, Germany; rolf.maier@med.uni-marburg.de; 2Department of Clinical Psychology and Psychotherapy, Philipps University of Marburg, 35032 Marburg, Germany; blanckep@staff.uni-marburg.de; 3Faculty of Medicine, Deans Office, Research Group Medical Ethics, Philipps University of Marburg, 35033 Marburg, Germany; carola.seifart@staff.uni-marburg.de

**Keywords:** breaking bad news, communication, parent–physician relationship, educational level, NICU, preterm

## Abstract

Communication in neonatal intensive care units and the relationship between families and staff have been reported to influence parental mental well-being. Research has also shown an impact of parental educational level on their well-being. However, whether different educational levels result in different reactions to breaking bad news (BBN) by physicians remains unanswered so far. We therefore examined the impact of parental level of education on their mental state after a BBN conversation and their relation to physicians. A prospective quantitative survey was conducted amongst 54 parents whose preterm or term infants were hospitalized in three German neonatal units. Parental education was classified as low (lower secondary/less (1), *n*: 23) or high (higher secondary/more (2), *n*: 31). Parents answered questions about certain aspects of and their mental state after BBN and their trust in physicians. The two groups did not differ significantly in their mental condition after BBN, with both reporting high levels of exhaustion and worries, each (median (min;max): (1): 16 (6;20) vs. (2): 14 (5;20), (scaling: 5–20)). However, lower-educated parents reported a lower trust in physicians (median (min;max): (1): 2 (0;9) vs. (2): 1 (0;6), *p* < 0.05 (scaling: 0–10)) and felt less safe during BBN (median (min;max): (1): 15 (9;35) vs. (2): 13 (9;33), *p* < 0.05). Only among higher-educated parents was trust in physicians significantly correlated with the safety and orientation provided during BBN (r: 0.583, *p* < 0.05, r: 0.584, *p* < 0.01). Concurrently, only among less-educated parents was safety correlated with the hope conveyed during BBN (r: 0.763, *p* < 0.01). Therefore, in BBN discussions with less-educated parents, physicians should focus more on giving them hope to promote safety.

## 1. Introduction

Premature birth and congenital malformations or diseases in newborn babies are an inherent and recurrent aspect of human life. Globally, approximately one in ten infants is born prematurely [1] and one in six newborns requires further medical support after birth [2]. Consequently, admission to a neonatal intensive care unit (NICU) becomes necessary to provide appropriate medical care, not seldomly resulting in the separation of the newborn from her/his parents.

Prematurity carries a significant risk of serious health problems for the newborn [3,4]. A newborn’s stay in a NICU can induce stress in parents, leading to various psychological symptoms such as depression, anxiety, or even post-traumatic stress symptoms [5,6,7]. Stress and worry increase with the occurrence of complications or illness of the infant.

In addition, further factors influence parents’ experience of stress and well-being, including their personal characteristics [8], social support [9], and their relationships and communication with nurses and physicians [10,11,12], which may be characterized by trust or lack thereof [9,13].

When medical problems arise, physicians have to inform parents about the situation. Despite the negative content, an empathic, patient-/recipient-centered way of communication with room for hope can sustain patients/relatives [14,15,16,17]. BBN is a challenging task for physicians and can affect the physician–parent relationship, with an insensitive communication of diagnoses having a negative impact [18]. This underscores the importance of guidelines for BBN for professionals. SPIKES is the only widely taught protocol [16], suggesting an orientation toward the needs of the recipient. However, existing protocols do not provide specific recommendations for BBN to different groups of recipients, e.g., patients with lower educational levels compared to patients with higher educational levels [15,16,18].

This is a gap, as Seifart et al. [19] demonstrated the influence of educational level on wishes and preferences regarding BBN. In addition, Baird [20] and Enke [12] reported effects of various parental- or child-related variables on stress or satisfaction in BBN in pediatrics too. In general, studies state an influence of recipient variables such as the socioeconomic status, measured by educational level, on health behaviors per se [21,22,23] as well as parenting stress [24].

Specifically, in neonatology, the influence of parental socioeconomic status on various aspects is well documented: it can impact the type of therapeutic decisions parents make, neonatal morbidity, and the long-term developmental outcome of preterm infants [3,25,26,27,28,29,30]. Therefore, socioeconomic status seems to play an important role in NICUs whereat an infant´s health risks generally increase with the lower educational status of the parents [3,25,26,27,28,29,30]. Additionally, factors such as parents´ level of education, their age, and social support have been found to influence the level of stress experienced in NICUs [31,32]. Nevertheless, to our knowledge, no surveys have been conducted to investigate the influence of educational level on stress, well-being, and trust in physicians in NICUs in relation to BBN (Box 1).

Box 1Research in Context.
**Research in context**

**Evidence before this study**
We started searching PubMed between September and November 2018 before submitting the study to the institutional review board of the Philipps University of Marburg. We updated our PubMed search on 6 December 2022 and 29 December  2022, while preparing and before submitting this manuscript, more than four years after we had
registered our study. Our search terms were the following:(((((parent[All Fields] OR parent’s[All Fields] OR parental[All Fields] OR parenthood[All Fields] OR parenting[All Fields] OR parents[All Fields] OR parents’[All Fields]) OR (relative[All Fields] OR relatives[All Fields]) OR (““infant, newborn””[MeSH Terms] OR newborn[Text Word]) OR (neonatal[All Fields] OR neonatale[All Fields] OR neonatally[All Fields] OR neonate[All Fields] OR neonates[All Fields] OR neonati[All Fields] OR
neonatologist[All Fields] OR neonatologists[All Fields] OR neonatology[All Fields] OR neonatus[All Fields]) OR (““premature birth””[MeSH Terms] OR preterm[Text Word]) OR (““intensive care units, neonatal””[MeSH Terms] OR NICU[Text Word]))) AND (breaking bad new* OR bad new* deliver* OR SPIKES OR communication OR doctor OR pediatr* OR paediatr*)) AND (educat* OR income OR
education level)) AND (stress OR distress OR trust OR satisfaction OR well-being OR mental health OR depress OR anxiety OR health)”.We identified no studies focusing on mental well-being in parents of newborns in a neonatal intensive care unit after having been delivered bad news, searching for differences in the level of education. As outlined in the introduction and discussion, we did find a few studies related to the present research in that the same target population was focused. However, comparators differed as well as endpoints. 
**Added value of this study**
This analysis of prospectively collected data revealed for the first time that parents with a lower educational status reported less trust in attending/treating physicians than those with a higher educational level and that physicians failed to communicate safety for lower-educated parents. It was found that only among higher-educated parents were the studied parameters of the BBN conversation correlated with trust in physicians. Moreover, only among parents with lower levels of education did hope and security conveyed in a BBN conversation correlate strongly with each
other.

In previous analyses we defined determinants influencing BBN in neonatology [33] and compared parental preferences with their actual BBN experiences [34] and found that BBN not only burdened the recipients but also the messengers. The majority of parental top preferences differed significantly in their implementation in the real-life interview, and a compassionate delivery of bad news correlated highest with a parent-rated quality of BBN conversation. Based on this work and the mentioned influence of parental educational level in neonatology on various aspects, we wanted to know whether parents respond differently to BBN according to their educational level. Additionally, we wanted to know whether there was a relationship between the parent–physician relationship and the parental sense of safety and orientation elicited by the physician’s BBN interview. This connection is based on Grawe’s consistency theory, which names basic human needs, such as self-efficacy, safety, and orientation. It can be used to explain the reactions of people in exceptional psychological situations [35,36] and helps to derive communication strategies for them [37].

## 2. Materials and Methods

### 2.1. Ethics

The institutional review board of the Philipps University of Marburg approved this study (ID-No.: 164/18). It was conducted according to the Helsinki Declaration, Good Clinical Practice guidelines, and the EU data collection directive. Parents were informed about the study before participation in written form. They consented by completing the questionnaire.

### 2.2. Participants and Procedure

This study was conducted in identical paper/pencil and online versions. From December 2018 to September 2020, data were collected prospectively in three neonatal units: the University Hospital Marburg, the Carl-Thiem-University-Hospital Cottbus (each highest care level) and the Hospital of Bad Hersfeld (middle care level). Usually, both parents received one questionnaire each and were approached by the researchers after bad news concerning their infant had been broken to them. Additionally, a freely accessible QR code, to be found on the web page of the German federation ‘The preterm Infant’, gave access to the online version. Inclusion criteria contained a serious diagnosis of the infant, such as serious congenital malformation, intraventricular hemorrhage, pneumothorax, or necrotizing enterocolitis, about which neonatologists or pediatric surgeons informed the parents in terms of delivering bad news. Usually, only attending physicians broke bad news. However, information about the level of education of the physician and the type of residency training was not collected. Participants were asked to refer to the first time bad news had been broken to them. The conversation should not have been more than six weeks ago. Questionnaires returned after this period were nevertheless accepted. Usually, the questionnaire was answered within two weeks after bad news’ delivery. Sufficient German language skills were required, as well as a minimum age of 18 years. Exclusion criteria were an age of less than 18 years, insufficient German language, and lack of consent to participate in the study.

### 2.3. Demographic Characteristics

Twenty-two items collected demographic data. These included age, sex, marital and educational status of the responding parent, complications during pregnancy, illness or complication of the infant, gestational age at time of birth, and parental evaluation of the infant’s current health condition (see Table 1).

### 2.4. Parental Mental Condition, Self-Efficacy, and Trust as well as Certain Aspects of BBN

The questionnaire asked for the procedure and perception of the first BBN consultation as well as preferences for BBN additionally to the quality of BBN in terms of satisfaction with the conversation, the condition after bad news’ delivery, self-efficacy, and their relationship with the treating physicians. The development and structure of the questionnaire as well as results concerning parental experiences, preferences, and satisfaction with BBN are presented elsewhere [34].

In contrast to that first report, we now focus on the following parameters: parents’ mental state after a BBN conversation, their self-efficacy, their assessment of certain behaviors of physicians in the BBN conversation that convey either ‘safety’ or ‘orientation’, and their relationship with the physicians as a function of parental educational level.

To survey the parental condition past BBN, five questions asked for the condition in the first few days after BBN, concerning aspects like sleep, concentration, worry, or exhaustion (for further information, see Appendix A Table A1 and Table A2).

To check for the mental condition, we used the following questionnaires: The PHQ-4 (Patient Health Questionnaire-4 [38]), the PSS-10 (Perceived Stress Scale-10 [39,40]), one question about current satisfaction with physical and mental well-being, and one for the current emotional burden/stress of the ‘present situation’. Four questions were asked for the overall self-estimation of self-efficacy. One item referred to the current relationship with physicians in terms of trust (for further information, see Appendix A Table A1 and Table A2).

Based on Grawe’s consistency theory, nine and eight options, respectively, were used to query whether certain behaviors of physicians in the BBN interview elicited ‘safety’ or ‘orientation’. Parents were asked to rate nine or eight statements, respectively, which followed the following phrase: ‘Physician conveyed safety/orientation in the conversation in that he/she…’ (for further information, see Appendix A Table A1 and Table A3). Additionally, in the section asking for parental experiences with their first BBN interview (see [34]), parents were asked to respond to singular statements that were rated on a four-point Likert scale (1 = ‘entirely’ to 4 = ‘not at all’). Four of them corresponded to ‘safety’ and ‘orientation’ too and were therefore also evaluated (see Section 3.4). In our former report, the five items related to the parental experience of the BBN conversation with the lowest agreement rates were listed. Two of the items identified in this way related to the communication of ‘hope’ and ‘safety’ too [34].

Seven items asked about certain characteristics of the setting and physician.

### 2.5. Data Analysis

Low parental education (lower secondary education or less in accordance with the International Standard Classification of Education, revised 2011 (ISCED-2011)) was compared with higher secondary education or more (according to Sentenac et al. [25]) for all reported data. Descriptive data were calculated with median and range, and mean and standard deviation for PSS-10, due to available validating data [40]. For the PHQ-4, the valid percentage of subjects who exceeded the clinical cut-off was calculated. Differences in demographic data according to parental educational level were calculated with Chi^2^ and Mann–Whitney U tests. To examine the difference for certain characteristics of the BBN interview, in the reported condition after BBN and trust in physicians according to the level of education, Mann–Whitney U tests were calculated. Cohen’s d was calculated as an effect size measure [41,42]. To examine the relationship between trust in physicians and medical behavior during BBN that elicited either safety or orientation, correlation coefficients (Spearman) were calculated. Fisher’s Z was used to test the differences in correlations between the two groups for significance. Comparatively, simple linear regressions were also calculated to assess the influence of educational status. To check for internal consistency, Cronbach’s alpha was calculated for items referring to either eliciting safety or orientation (see Appendix A Table A3).

## 3. Results

### 3.1. Sample and Demographic Data

Fifty-four questionnaires were returned altogether, seven of which were in the online version, with 23 parents with a lower secondary education or less (group 1) and 31 parents with an upper secondary education or more (group 2). The demographic and medical characteristics of the parents and infants are listed in Table 1. Most of the respondents were mothers ((group 1): 87%, (group 2): 81%) and were between 30–39 years old ((group 1): 65%, (group 2): 77%). Antenatal fetal problems were reported by 26% (group 1) versus 13% (group 2) of the respondents. Nine percent (group 1) versus 19% (group 2) stated a twin pregnancy. The rate of complications during pregnancy was high for both groups ((group 1): 44%, (group 2): 52%). In group 1, 43% of the infants were born mature and in group 2, 29% were. None of the demographic data differed significantly between the two groups. The survey was anonymized. Therefore, we could not match the physician’s diagnoses with information of the parents about their newborn’s disease and their evaluation of the infant’s health status and not specify the level of care of the corresponding hospitals. The parental responses to diagnoses were too imprecise for further analysis. We were unable to verify whether the seven participants who completed the questionnaire online actually had an infant with a serious diagnosis. However, we asked for the type of complication/disease, and answers seemed plausible, such as severe asphyxia or intraventricular hemorrhage.

### 3.2. Mental Condition after the Delivery of Bad News

The mental condition for a lower versus higher educational status is listed in Table 2. Looking at the valid percentage in group 1, 19% exceeded the clinical cut-off for depression (>3) and 20% in group 2 did, with no significant difference between the two groups (U: 301.00, Z: −0.273, *p*: 0.785). For anxiety, 28.6% of persons in group 1 exceeded the clinical cut-off (>3) and 21.4% in group 2 did, with no significant difference between the groups either (U: 293.50, Z: −0.010, *p*: 0.992). This incidence was higher compared to the general population [43,44]. Both groups reported a high burden in terms of exhaustion, sleeplessness, or worries the first few days after BBN (’well-being and condition after BBN’: median (min;max): (group 1): 16.00 (6;20), (group 2): 14.00 (5;20)). Looking at Cohen’s d, we found only small effect sizes. Our comparatively small sample size makes it difficult to actually detect differences between the two educational groups.

### 3.3. Stress, Self-Efficacy, Helplessness, and Life Satisfaction

The results for the PSS-10, current self-estimated stress, current satisfaction with physical and mental well-being, and overall self-estimation of self-efficacy, are listed in Table 2 as well. All parents reported a high ‘perceived helplessness’ (PSS-10: mean (sd): (group 1): 13.67 (3.864), (group 2): 14.36 (5.539)), exceeding the means of the German comparison group by more than one standard deviation (for values of the comparison group, see [40]). Self-estimated stress and self-efficacy as well as satisfaction with well-being were in a medium range each (stress: median (min;max): (group 1): 5.00 (1;10), (group 2): 5.50 (1;10)/self-efficacy: median (min;max): (group 1): 9.00 (5;12), (group 2): 8.00 (4;12)/satisfaction with well-being: median (min;max): (group 1): 2.50 (1;4), (group 2): 2.00 (1;4)).

### 3.4. Certain Aspects of the BBN Conversation and Trust in Physicians

Results for current trust in physicians and physicians’ behavior in the BBN interview that elicited safety or orientation are listed in Table 3. Responding parents with a lower educational status reported significantly less trust in physicians (U: 244.50, Z: −2.035, *p*: 0.042, d: 0.598) than did those with a higher educational status. For both the nine items that asked about a particular physician’s behavior, which was supposed to provide reassurance (U: 196.50, Z: −2.136, *p*: 0.033, d: 0.513) and the single item ‘Physician gave me safety’ (U: 190.00, Z: −2.188, *p*: 0.029, d: 0.654), less-educated parents reported that physicians failed to make them feel safe. There was a tendency, though not significantly, for less-educated parents to not have sufficient opportunities for questions, and they felt that physicians transported only little hope. Despite our small sample size, effect size measures were predominantly found in a medium range (see Table 3), which supports this being a real effect, detectable even in a small sample.

### 3.5. Correlation of Safety, Orientation, Certain Characteristics of the BBN Conversation, and Trust in Physicians

To check if the BBN conversation elicited safety and orientation among parents and if this was related to trust in physicians, Spearman correlation coefficients were calculated (see Table 4). A high transmission of safety went along with a high transmission of orientation for both groups ((group 1): r: 0.784, (group 2): r: 0.848, with *p* < 0.01 each). In both groups, the ability to ask questions correlated significantly with safety and orientation (see Table 4), whereby the correlation between asking questions and orientation tended to be higher for less-educated parents ((group 1): r: 0.820, (group 2): r: 0.508, Z: 1.864, *p*: 0.062). Only among higher-educated parents did we find significant correlations between trust in physicians and safety (‘item safety’: (group 1): r: 0.198, (group 2): r: 0.715, Z: −2.2538, *p*: 0.024) and understanding ((group 1): r: −0.036, (group 2): r: 0.493, Z: −1.9675, *p*: 0.049), with significant differences of the correlation coefficients between the two groups. Additionally, only among less-educated parents did hope correlate with safety (‘item safety’: (group 1): r: 0.763, (group 2): r: 0.208, Z: 2.5630, *p*: 0.01). The correlation between hope and ‘asking questions’ tended to be higher for less-educated parents as well ((group 1): r: 0.588, (group 2): r: 0.171, Z: 1.6237, *p* = 0.104). A linear regression analysis supported Fisher’s Z results.

## 4. Discussion

In our investigation of the differences in the mental reactions in relation to BBN conversations and the relationship with physicians based on parental education level, we observed that lower-educated parents reported less trust in physicians.

In addition, these parents felt less secure from the BBN conversation than did parents with higher levels of education. They also tended to say that physicians did not give them enough hope and they did not have enough opportunities to ask questions. We found no differences in mental condition in comparison, due to the different parental levels of education. The rate of those reporting elevated levels of depression and anxiety was higher than that of the general population in both groups. What stood out was the significantly increased ‘perceived helplessness’ for both groups compared to the norm sample. The transmission of safety and orientation through the BBN conversation as well as asking questions correlated highly for either educational group. Only among higher-educated parents did we find a correlation between safety and trust in physicians and an understanding of the content and trust. For less-educated parents, we did not find any correlation between the different aspects of the BBN conversation and their confidence. However, for them, the communication of hope seemed to be of greater importance, as hope here correlated strongly with safety and the opportunity to ask questions.

We can assume that less-educated parents have different needs for support to develop trust and may feel uncomfortable in a conversation with highly skilled physicians, perceiving an imbalance of power. These findings align with October et al. [45], who found that language disparities between healthcare teams and families in intensive care unit conferences could hinder family engagement and satisfaction with BBN. Limiting medical jargon could improve communication and satisfaction with BBN [45,46,47]. According to Rudd and Katz, lower (health) literacy also impairs understanding [48] and is associated with a passive communication style with physicians in which fewer or irrelevant questions are asked, challenging physicians to engage their counterparts as actively as possible using various conversational techniques [49]. In addition, higher educational levels may be associated with “similar world views about medical practice” (p. 11) [50] between parents and physicians, leading to greater trust, as reported by Krupat [50].

A look at the results of the correlation calculations can additionally complete the picture. Based on Grawe’s consistency theory, we initially hypothesized that specific aspects of the BBN conversation would influence the physician–parent relationship. However, we only observed this connection among higher-educated parents, with strong correlations between trust and a sense of safety and orientation. Apparently, for parents with lower levels of education, the BBN interview has a lesser effect on the relationship with the physicians than it does for higher-educated parents. Krupat found that patients with higher levels of education have a stronger preference for receiving information during medical consultations and thus develop a greater sense of control [50]. This, in turn, may affect trust in the interlocutor, as well as feelings of safety, and fits with our finding that better-educated parents reported feeling significantly safer as a result of the BBN interview. Another explanation may be that clinicians simply failed to create a sense of security in parents with lower levels of education during the BBN interview, resulting in a lack of correlation with confidence.

Even though for both groups the opportunity to ask questions correlated strongly with the safety and orientation provided by physicians, the less educated tendentially showed a stronger correlation between asking questions and orientation. Moreover, for less-educated parents, the ability to ask questions additionally correlated strongly with the hope conveyed. This once again underscores the need to work harder to ensure that less-educated parents in particular actively participate in the conversation and actually ask their questions [49]. For parents with lower levels of education, this seems to be even more important than actually understanding, which underscores the importance of this simple communication technique for achieving higher-level, desired goals such as hope, security, and orientation among parents. Hope, which is often highlighted as one of the main goals of successful BBN in general [15,51], only correlated with safety and orientation among less-educated parents. On the other hand, only among higher-educated parents did an understanding of the BBN content correlate with safety and orientation. In summary, then, in addition to encouraging questions, physicians should focus more on addressing the emotions in the case of parents with lower levels of education and on addressing their need for understanding in the case of parents with higher levels of education. For a brief overview of our recommendations for BBN in NICUs based on our findings, see Box 2).

The topic of hope is worth taking a closer look at. Undoubtedly, a type of diagnosis and its resulting prognosis are related to hope. Unfortunately, due to anonymous data collection, we were unable to validly relate an infant’s diagnosis as a confounding variable to parental responses about whether physicians conveyed hope. However, for the ‘demographic’ data, we found no significant difference in the parental assessment of the infant’s prognosis between the two groups (see Table 1), although tendentially, only lower-educated parents reported that physicians did not give them enough hope. There is evidence in the literature that the understanding of hope varies between physicians and parents, which reduces trust in physicians, complicates communication between each other, and may offer an explanation for why parents indicated that little hope was conveyed in the BBN conversation [52]. For parents, hope functions as an emotional motivator [52] and arises from the opening of a future perspective of whatever kind [53] and by focusing on the positive [54]. This must not be confused with a hope for healing that is far removed from reality, as is sometimes perceived by physicians [52].

Regarding physical and psychological well-being, both groups of parents reported similarly high levels of stress in the week following BBN and of depression and anxiety. So, we dealt with highly burdened parents but to a lesser extent than reported by Soghier et al. [32], who found levels of depression in NICU parents by up to 45%, possibly due to the different survey tools. Younge et al. [31] also found differences in stress scores among parents with different educational levels in NICUs, which may be due to the larger sample and a broader range of stress-related parameters that were collected. The detected differences only appeared in relation to parental and NICU-elicited stress parameters but not in the PSS-10. Nevertheless, the consistently high stress levels, mainly due to high reported helplessness, remind NICU staff of the importance of highly empathetic communication during and after BBN discussions, as highlighted by Enke et al. [12]. Regarding anxiety and depression, it should be noted that we collected the data shortly after BBN. To our knowledge, there are no data on the temporal evolution of anxiety and depression directly related to BBN in NICUs. However, data on postnatal anxiety and depression in parents in general show a decrease in psychological symptoms over time [55,56]. At the same time, socioeconomic status is reported to influence the likelihood of occurrence of anxiety and depression in the way that mothers with higher levels of education are at a lower risk for it. In this respect, perhaps a larger sample would have led to a somewhat different picture.

The high reported helplessness of all parents deserves special attention, since helplessness can indicate that basic psychological human needs, as stated by Grawe, may not have been met [35,36]. This is problematic, because a low satisfaction of needs makes it difficult to cope well with challenging borderline situations (e.g., [57]) and to provide good, supportive care for one’s own infant [9,58,59]. An infant’s stay in a NICU in itself is often a highly stressful situation for parents [60,61], even without bad news about newly diagnosed diseases, and helplessness-based liminality [62] shapes parental NICU experiences anyway. Thus, it is indispensable for the entire health care team to orient its interaction with parents so that they can experience and behave in a self-efficient and competent manner [11,58] and reduce helplessness by actively integrating parents in their daily care of their infants [10,63]. Related to the delivery of bad news, Fallowfield broadens the view to the entire NICU team in emphasizing that too often a BBN conversation is reduced to one informing physician while the surrounding team, with its ongoing communication before and after BBN discussions, is underappreciated and overlooked in their influence of parents [18].

There are some limitations that should be noted. Since medical complications of preterms may as well occur after the family has already spent weeks in a NICU, a trustful relationship may have already been built up, independent of the first breaking bad news conversation. Future studies should control for this in asking for trust in physicians before and after bad news has been broken or if BBN influenced trust in physicians at all from the parent’s point of view. In addition, we did not ask about mediating or moderating variables that influence the parameters we queried other than educational status or a conversation about bad news, such as general social support, certain attitudes, response tendencies, medical factors such as the number of pregnancies, or others. To control for this, the study population needs to be larger too. Moreover, more fathers should be included as well as parents with a primary educational level or no graduation to better represent this group. In general, future studies should include a larger sample to generalize results more validly.

Box 2Recommendation box.
**Recommendation box for the delivery of bad news in the NICU to evoke ‘safety’ and ‘orientation’ in parents**
Limit medical jargon in BBN conversations wherever possible and use plain ‘everyday’ language.Actively encourage parents to ask questions (i.e., “For many parents questions arise in such a situation. What are yours?”), especially lower-educated parents.Focus more on addressing emotions among lower-educated parents and especially convey hope, e.g., by creating a future perspective.In contrast, among higher-educated parents, focus more on explanations and their understanding of the content


## 5. Conclusions

Higher-educated parents generally reported more trust in physicians and a higher sense of safety elicited by the BBN conversation than less-educated parents. Therefore, future research should focus specifically on what factors might be important in developing trust for parents with lower levels of education. The ability to ask questions, which is highly correlated with safety and orientation, appeared to be vital for all parents, so physicians should pay particular attention to encouraging parents to ask their questions. Only among less-educated parents does safety correlate highly with hope, which is why physicians should take greater care to convey hope when delivering bad news to them. Because of the high level of parental helplessness, regardless of educational level, the general interaction of all NICU staff with parents should aim to reduce their helplessness in order to reduce symptoms of stress and thus strengthen their parenting. In the sense of stepwise research, the present exploratory work may lay a foundation for subsequent experimental studies on the delivery of bad news in neonatology.

## Figures and Tables

**Table 1 children-10-01729-t001:** Demographic and medical characteristics (total *n* = 54).

					Test Statistics					
	1: Lower Secondary Education or Less ^a^		2: Upper Secondary Education or More ^a^		χ^2^ (1) ^b^	*p*-Value	Phi	U	Z	*p*-Value ^c^
	*n*	%	*n*	%						
**Parental characteristics**	23	43	31	57						
Age (years) (md (25th/75th p.)	32 (29/35)		34 (32/36)					251.00	−1.843	0.065
20–29	7	30	4	13						
30–39	15	65	24	77						
40–49	0	0	3	10						
50–59	1	5	0	0						
Sex					FeT ^d^	0.717	0.084			
Father	3	13	6	19						
Mother	20	87	25	81						
Marital Status					FeT ^d^	0.569	0.118			
Single	2	9	1	3						
Married/living with partner	21	91	30	97						
Citizenship					FeT ^d^	0.380	0.146			
German	22	96	27	87						
Other	1	4	4	13						
Pregnancy and delivery										
Artificial fertilization (not specified)	4	17	2 (1)	7 (3)	FeT ^d^	0.385	0.146			
Antenatal fetal problems (not specified)	6	26	4 (2)	13 (7)	FeT ^d^	0.307	0.155			
Twin pregnancy (not specified)	2	9	6 (1)	19 (3)	FeT ^d^	0.443	0.195			
Complication during pregnancy (not specified)	10	44	16 (1)	52 (3)	0.506	0.477/0.583	0.098			
Complication during delivery (not specified)	4 (2)	19 (9)	8 (2)	26 (7)	FeT ^d^	0.517	0.110			
**Gestational age (GA) at time of birth**										
GA (weeks (25th/75th p.))	33.5 (24.8/37.3)		30.0 (26.0/39.5)					303.0	−0.098	0.922
≤27+6 (very early preterm)	9	39	13	42						
28+0–31+6 (early preterm)	0	0	3	10						
32–36+6 (preterm)	3	13	3	10						
>37+0 (mature born)	10	43	9	29						
Not specified	1	4	3	10						
**Condition of infant at time of answering the questionnaire according to parental estimation**								341.5	−0.301	0.727
Very well/well	19	83	22	71						
Moderate	3	13	5	16						
Child died	0	0	2	7						
Not specified	1	4	2	7						
**Parents’ self-estimated prognosis at time of answering the questionnaire**								344.5	−0.443	0.684
Very well/well	17	74	24	77						
Moderate/bad	5	22	4	13						
Not specified	1	4	3	10						

Note. ^a^ According to ISCED-2011. ^b^ χ^2^(df): degree of freedom. ^c^ Exact two-tailed. ^d^ FeT: Fishers exact test due to expected cell sizes of <5.

**Table 2 children-10-01729-t002:** Descriptive statistics and group differences for clinical data and self-efficacy according to different levels of education.

Scale/Item	Scale			MD/% ^b^	Min;Max	Test Statistics			
	Range	Group ^a^	*n*	(Mean)	(sd)	U	Z ^c^	*p*	d ^d^
Sum well-being and condition first few days after BBN low values = good well-being	5–20	1	23	16.00	6;20	325.50	−0.545	0.586	0.138
		2	31	14.00	5;20				
PHQ-4 depression low values = no depressive symptoms	0–6	1	21	19% ^b^	0;4	301.00	−0.273	0.785	0.012
		2	30	20% ^b^	0;6				
PHQ-4 anxiety low values = no symptoms of anxiety	0–6	1	21	28.6% ^b^	0;4	293.50	−0.010	0.992	−0.088
		2	28	21.4% ^b^	0;6				
PSS-10 perceived self-efficacy low values = high self-efficacy	0–16	1	21	8.00 (7.43)	1;12 (2.749)	263.00	−0.630	0.529	0.288
		2	28	6.00 (6.68)	0;13 (3.642)				
PSS-10 perceived helplessness low values = little helplessness	0–24	1	21	13.00 (13.67 ^e^)	8;22 (3.864)	276.00	−0.365	0.715	−0.141
		2	28	14.50 (14.36 ^e^)	4;24 (5.539)				
Current satisfaction with physical and mental well-being low values = high satisfaction with well-being	1–4	1	20	2.50	1;4	238.50	−0.749	0.454	0.181
		2	27	2.00	1;4				
Current self-estimated stress caused by situation/emotional burden low values = no stress	0–10	1	22	5.00	1;10	325.00	−0.093	0.926	0.026
		2	30	5.50	1;10				
Sum overall self-estimation of self-efficacy low values = high self-efficay	4–16	1	22	9.00	5;12	315.00	−0.077	0.939	−0.009
		2	29	8.00	4;12				

Note. ^a^ 1: lower secondary education and less according to ISCED, revised 2011; 2: upper secondary education and higher according to ISCED, revised 2011. ^b^ Valid percentage exceeding the clinical cut-off of >3 for PHQ-4 depression and anxiety. ^c^ Mann–Whitney U test. ^d^ Cohen’s d for effect size. ^e^ >1sd higher than the mean of reference group (see [40]).

**Table 3 children-10-01729-t003:** Descriptive statistics and group differences for certain characteristics of the BBN conversation such as safety and orientation/knowledge and the relation to physicians according to different levels of education.

Scale/Item	Scale			MD	Min;Max	Test Statistics			
Certain Aspects of BBN Conversation	Range	Group ^a^	*n*			U	Z ^b^	*p*	d ^c^
‘I understood all explanations’ (orientation) ^d^	1–4	1	23	2.00	1;4	302.00	−1.044	0.296	0.296
		2	31	1.00	1;4				
‘I had sufficient opportunity for questions’ (orientation) ^d^	1–4	1	22	1.50	1;4	238.00	−1.858	0.063	0.331
		2	29	1.00	1;4				
‘Physician transported hope’ (safety) ^d^	1–4	1	21	2.00	1;4	211.00	−1.760	0.078	0.562
		2	28	2.00	1;4				
‘Physician gave me safety’ (safety) ^d^	1–4	1	21	3.00	1;4	190.00	−2.188	0.029 *	0.654
		2	28	2.00	1;4				
Sum ‘safety’ low values = physicians conveyed a lot of safety	9–36	1	21	15.00	9;35	196.50	−2.136	0.033 *	0.513
		2	29	13.00	9;33				
Sum ‘orientation’ ^e^ low values = physicians gave a lot of orientation	7–28	1	19	18.00	8;27	201.50	−1.402	0.161	0.399
		2	28	14.50	7;27				
Relationship with physicians (trust) low values = a lot of trust	0–10	1	23	2.00	0;9	244.50	−2.035	0.042 *	0.598
		2	31	1.00	0;6				

Note. ^a^ 1: lower secondary education and less according to ISCED, revised 2011; 2: upper secondary education and higher according to ISCED, revised 2011. ^b^ Mann–Whitney U test. ^c^ Cohen’s d. ^d^ (1: agree ‘entirely’, 4: agree ‘not at all’). ^e^ Due to low Cronbach’s alpha exclusion of one item for ‘orientation’ (see Appendix A Table A3). * *p* < 0.05.

**Table 4 children-10-01729-t004:** Correlation coefficients for certain characteristics of the BBN conversation concerning safety and orientation, and relation to physicians according to different levels of education.

		Understanding		Asking Questions		Hope		Item Safety		Sum ‘Safety’		Sum ‘Orientation’	
	Group ^a^	1	2	1	2	1	2	1	2	1	2	1	2
Asking questions	R ^b^ (N)	0.321 (22)	0.373 * (29)	__	__	__	__	__	__	__	__	__	__
	Z^c^ (*p* ^d^)	−0.196 (0.845)											
Hope	R ^b^ (N)	0.048 (21)	−0.005 (28)	0.588 ** (21)	0.171 (28)	__	__	__	__	__	__	__	__
	Z ^c^ (*p* ^d^)	0.1716 (0.864)		1.6237 (0.104)									
Item safety	R ^b^ (N)	0.074 (21)	0.467 * (28)	0.560 ** (21)	0.579 ** (28)	0.763 ** (21)	0.208 (28)	__	__	__	__	__	__
	Z ^c^ (*p* ^d^)	−1.3978 (0.162)		−0.0910 (0.928)		2.5630 (0.01 **)							
Sum ‘safety’	R ^b^ (N)	0.286 (21)	0.140 (29)	0.603 ** (21)	0.390 * (29)	0.598 ** (20)	0.194 (28)	0.702 ** (20)	0.710 ** (28)	__	__	__	__
	Z ^c^ (*p* ^d^)	0.4999 (0.617)		0.9329 (0.351)		1.5700 (0.116)		−0.0507 (0.960)					
Sum ‘orientation’	R ^b^ (N)	0.493 * (19)	0.172 (28)	0.820 ** (19)	0.508 ** (28)	0.492 * (18)	0.222 (27)	0.659 ** (18)	0.683 ** (27)	0.784 ** (19)	0.848 ** (28)	__	__
	Z ^c^ (*p* ^d^)	1.1441 (0.253)		1.8640 (0.062)		0.9508 (0.342)		0.1716 (0.864)		−0.1327 (0.894)			
Relation (Trust)	R ^b^ (N)	−0.036 (23)	0.493 ** (31)	0.175 (22)	0.344 (29)	0.130 (21)	−0.081 (28)	0.198 (21)	0.715 ** (28)	0.125 (21)	0.583 ** (29)	0.187 (19)	0.584 ** (28)
	Z ^c^ (*p* ^d^)	−1.9675 (0.049 *)		−0.6024 (0.547)		0.6856 (0.493)		−2.2538 (0.024 *)		−1.7655 (0.077)		−1.4970 (0.134)	

Note. ^a^ 1: lower secondary education and less according to ISCED, revised 2011; 2: upper secondary education and higher according to ISCED, revised 2011. ^b^ Spearmon rho. ^c^ Fisher’s Z. ^d^
*p*-value for Fisher’s Z (two-tailed). * *p* < 0.05. ** *p* < 0.01 (two-tailed).

## Data Availability

The data presented in this study are available on request from the corresponding author. The data are not publicly available out of ethical reasons and respect to participating families.

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
