# Peer review of "Does Educational Status Influence Parents’ Response to Bad News in the NICU?"

_children, 2023, doi:10.3390/children10111729_

Round 1
Reviewer 1 Report
The authors of the manuscript entitled: “Does educational status influence parents’ response to bad news in the NICU?” have shown the importance of developing more trust and a higher sense of safety for parents with lower educational status when delivering bad news. They provide valuable recommendations for pediatricians and neonatologists when breaking bad news to parents at the NICU.
The manuscript is well written and results are clearly presented. It does seem that the authors have already published a recent paper of the same study, dividing the dataset into separate publications which is not always preferrable.
Some remarks
I have some difficulties with the use of ‘room for hope’ (line 51) or ‘physicians transported only little hope (line 216). Would this not be highly dependent on the diagnosis and the current health status of the child? In some cases there is indeed ‘less hope’ than in other cases and as a pediatrician/neonatologist you want to be honest and realistic, not spreading any ‘false hope’ as well. You were unable to match the pediatric diagnosis to the results of the questionnaire (as stated in line 170), but this could be a severe confounding factor. Could the authors specify this in the manuscript?
M&M. Line 99. Was this QR code available for everybody online? Was the questionnaire freely accessible? Were you able to confirm if parents who filled in the questionnaire online indeed had a serious diagnosis?
M&M. Were parents approached/selected by their treating pediatrician/neonatologist or by the researcher to participate in the questionnaire study? Could that lead to some sort of bias, as the pediatricians were the ones performing the BBN?
General remark about M&M, results: there is some overlap in presented data and text between this paper and your recently published paper: https://pubmed.ncbi.nlm.nih.gov/37336635/. The authors should specify the difference and overlap between the studies in more detail.
Line 192. In this small sample size, I would not mention this observed trend in your data as it is lacking supportive evidence.
Discussion. Line 256. There is no significant difference in mental condition between the different education groups two weeks after the bad news delivery. However, one could imagine this period is very hectic whatsoever. Do the authors expect a difference on the long term? Is any literature available on this topic?
Discussion. The authors pose some important recommendations for pediatricians/neonatologist when breaking bad news to parents with lower educational status in different paragraphs of the discussion. Could the authors include a recommendations box with bullet points as an overview?
Some minor comments
Line 52 ‘insensitive transmission of diagnoses’, vague use of language. Consider rephrasing.
Line 56 ‘younger versus older parents’ is used as an example here. Why not use the example ‘educational level’ as this is what your manuscripts entails?
Line 58. Please consider removing ‘for example’.
Line 65. ‘It’ should not be written with a capital letter
Text box: ‘transport safety’, would consider rephrasing.
Line 202. ‘to the comparison group’, please mention that this comparison group is a German community sample.
Line 297. I would suggest combining this and the next paragraph.
Reviewer 2 Report
This study ecaluated the impact of educational status on parents’ response to bad news in the NICU. It is an interesting paper. The authors should further clarify whether this is a retrospective study or a retrospective analysis of prospectively collected data. In addition, the authors should explain why the period between December 2018 and September 2020 was selected. Is the number of returned questionnaires big enough to draw any deffinite conclusions? What were the exclusion criteria?
